# Combination RSL3 Treatment Sensitizes Ferroptosis- and EGFR-Inhibition-Resistant HNSCCs to Cetuximab

**DOI:** 10.3390/ijms23169014

**Published:** 2022-08-12

**Authors:** Shujie Liu, Shuai Yan, Jie Zhu, Ruiqing Lu, Chujie Kang, Kang Tang, Jinfeng Zeng, Mingmei Ding, Zixiang Guo, Xianxin Lai, Yinan Jiang, Siqing Wu, Lihua Zhou, Litao Sun, Zhong-Wei Zhou

**Affiliations:** 1School of Medicine, Shenzhen Campus of Sun Yat-sen University, Shenzhen 518107, China; 2School of Public Health (Shenzhen), Shenzhen Campus of Sun Yat-sen University, Shenzhen 518107, China

**Keywords:** head and neck squamous cell carcinomas, ferroptosis, *FTH1*, cetuximab resistance

## Abstract

Head and neck squamous cell carcinomas (HNSCCs) are a type of cancer originating in the mucosal epithelium of the mouth, pharynx, and larynx, the sixth most common cancer in the world. However, there is no effective treatment for HNSCCs. More than 90% of HNSCCs overexpress epidermal growth factor receptors (EGFRs). Although small molecule inhibitors and monoclonal antibodies have been developed to target EGFRs, few EGFR-targeted therapeutics are approved for clinical use. Ferroptosis is a new kind of programmed death induced by the iron catalyzed excessive peroxidation of polyunsaturated fatty acids. A growing body of evidence suggests that ferroptosis plays a pivotal role in inhibiting the tumor process. However, whether and how ferroptosis-inducers (FINs) play roles in hindering HNSCCs are unclear. In this study, we analyzed the sensitivity of different HNSCCs to ferroptosis-inducers. We found that only tongue squamous cell carcinoma cells and laryngeal squamous cell carcinoma cells, but not nasopharyngeal carcinoma cells, actively respond to ferroptosis-inducers. The different sensitivities of HNSCC cells to ferroptosis induction may be attributed to the expression of *KRAS* and ferritin heavy chain (*FTH1*) since a high level of *FTH1* is associated with the poor prognostic survival of HNSCCs, but knocked down *FTH1* can promote HNSCC cell death. Excitingly, the ferroptosis-inducer RSL3 plays a synthetic role with EGFR monoclonal antibody *Cetuximab* to inhibit the survival of nasopharyngeal carcinoma cells (CNE-2), which are insensitive to both ferroptosis induction and EGFR inhibition due to a high level of *FTH1* and a low level of *EGFR*, respectively. Our findings prove that *FTH1* plays a vital role in ferroptosis resistance in HNSCCs and also provide clues to target HNSCCs resistant to ferroptosis induction and/or EGFR inhibition.

## 1. Introduction

Head and neck squamous cell carcinomas (HNSCCs) are the sixth most common cancer in the world. HNSCCs can originate in any tissue or organ in the head or neck except the eyes, brain, ears, thyroid, and esophagus, mainly in the mucosal epithelium of the mouth, pharynx, and larynx. Squamous cell carcinoma is the most common pathological type [1,2]. The incidence of HNSCCs varies from country to country. Men are generally at 2~4 times higher risk than women for developing HNSCCs. The morbidity ranks sixth, and the mortality ranks seventh in males, which is increasing yearly and is anticipated to increase by 30% (about 1.08 million/year) by 2030 [2,3]. HNSCCs are usually associated with exposure to tobacco-derived carcinogens and/or excessive drinking [4]. Human papillomavirus (HPV) infection is another risk contributing to the high prevalence of HNSCCs [2]. For example, 24.7% of HNSCC patients in China were reported associating with HPV infection [5]. Particularly, tumors in the oropharynx are directly related to persistent HPV infection [6], of which the primary subtype is HPV16, followed by HPV18 and other subtypes [7,8].

HNSCCs proliferate rapidly, which results in regional lymph node metastasis and poor prognosis [9]. Multimodal therapy, including surgery, radiotherapy, and chemoradiotherapy, is the primary treatment for HNSCCs [2,10]. Epidermal growth factor receptor (EGFR) is the receptor of epithelial growth factor (EGF) in cell proliferation and signal transduction. The EGFR signaling pathway plays a pivotal role in physiological processes such as cell growth, proliferation, and differentiation [11]. Mutations in *EGFR* will continuously activate the EGFR signaling pathway, causing abnormal cell proliferation. The abnormal expression of EGFR can be found in many solid tumors, which promotes the growth and development of tumors [11,12]. It has also become a hot topic in HNSCC diagnosis and treatment [10]. Currently, there are 12 drugs targeting EGFR on the market, including eight small molecule inhibitors and four monoclonal antibodies. Monoclonal antibodies include *Cetuximab*, *Nimotuzumab*, *Panitumumab*, and *Necitumumab*, while small molecular inhibitors are *Gefitinib*, *Erlotinib*, *Icotinib*, *Lapatinib*, *Afatinib*, *Osimertinib*, *Neratinib*, and *Pyrotinib* [13]. However, even with multiple treatments, only 40% of HNSCC patients have a 5-year survival period, and nearly half of patients will experience a relapse [14,15]. Therefore, drugs with high selectivity, low toxicity, and reverse resistance to radiotherapy and chemotherapy are badly needed to treat HNSCCs.

When looking for better anticancer small molecules, Stockwell’s team started a high-throughput screening in 2001 and discovered a series of compounds that can induce cell death in a unique way that distinguish from apoptosis and necrosis, which named “ferroptosis” [16,17]. Ferroptosis, a novel, non-apoptotic form of programmed cell death (PCD), is generally induced by iron accumulation and lipid peroxidation-mediated cell membrane damage [18]. Therefore, altering iron metabolism or inactivating the cellular antioxidant system can induce the ferroptosis process [19]. Iron metabolism is regulated by a series of genes, including iron importing factors, such as transferrin (TFR) and divalent metal transporter 1 (DMT1); exporting factor ferroportin (FPN); and free iron storing complex ferritin [19]. Ferritin consists of ferritin light chain (FTL) and heavy chain 1 (FTH1, or FTH) which function to convert redox-active Fe^2+^ to redox-inactive Fe^3+^ [20]. FTH1 plays a vital role in regulating ferroptosis cell death since ferroptosis-sensitive cells express a lower level of *FTH1* than ferroptosis-resistant ones, and the deletion of *FTH1* can induce ferroptosis [21,22]. In addition to iron metabolism, the glutathione peroxidase 4 (GPX4)-mediated antioxidant system also plays an essential role in controlling ferroptosis [19]. GPX4 functions to convert reduced glutathione (GSH) to oxidized glutathione (GSSG), therefore reducing lipid hydroperoxides [19]. The synthesis of GSH requires cystine uptake by the system Xc- complex, an amino acid transporter composed of two subunits, SLC7A11 and SLC3A2 [19]. Compounds, such as Erastin, by inhibiting the activity of system Xc-, therefore affect the synthesis of GSH and RSL3, which can directly inhibit the activity of GPX4, inducing ferroptosis by reducing antioxidant capacity [19,22]. On the other hand, ferroptosis cell death can be reversed by iron chelation and substances that can prevent the formation of lipid peroxides, such as ferrostatin-1 (Fer-1), liproxstatin-1 (Lip-1), and vitamin E [18,19].

The ferroptosis-inducers Erastin and RSL3 have been shown to play a synergetic lethal effect in cells that harbor *RAS* mutations [14,15]. RAS is a small GTPase that controls normal physiological processes such as gene expression, cell cycle, and membrane transport in intracellular signal transduction [23,24]. Oncogenes of the *RAS* family (*HRAS*, *NRAS*, and *KRAS*), which lead to the chronic activation of RAS, are the most commonly mutated in all human cancers [18,25]. Therefore, ferroptosis has become a new and exciting targeting strategy for different types of cancer [18,26]. Resistance is common in cancer therapy, and emerging evidence suggests that ferroptosis plays a pivotal role in overcoming therapy resistance [18,26]. For example, it has been reported that targeting GPX4 can reverse Temozolomide-resistant glioblastoma, Oxaliplatin-resistant colorectal cancer, and Gefitinib-resistant breast cancer [26,27,28,29]. However, the function of ferroptosis in HNSCCs is not well studied. Moreover, whether ferroptosis-inducers play a synergetic function with other drugs, such as inhibitors or antibodies against EGFR, in killing HNSCCs is unclear.

This study tested the sensitivity of different HNSCCs, including tongue squamous cell carcinoma, laryngeal squamous cell carcinoma, and nasopharyngeal carcinoma cells. We discovered that nasopharyngeal carcinoma cells, which express a low level of *KRAS* and a high level of *FT**H1*, are resistant to Erastin and RSL3. Additionally, nasopharyngeal carcinoma cells also express a lower level of *EGFR* and are, therefore, resistant to EGFR inhibition treatment. Combining RSL3 and EGFR antibody Cetuximab impairs synergistically the survival of nasopharyngeal carcinoma cells. These data not only reveal important profiles regarding the general effect of ferroptosis-inducers in HNSCCs but also provide a new targeting strategy for drug-resistant HNSCCs.

## 2. Results

### 2.1. HNSCC Cell Lines Respond Differentially to Ferroptosis-Inducers

To explore the sensitivity of different types HNSCCs in ferroptosis induction, we treated tongue squamous cell carcinoma (CAL33), laryngeal squamous cell carcinoma (AMC-HN-8 and TU686), and nasopharyngeal carcinoma cells (CNE-2, S18, and S26) with the ferroptosis-inducers Erastin, RSL3, or Sorafenib. We found that the cellular viability, measured by CCK8 assay, of tongue squamous carcinoma cells and laryngeal squamous carcinoma cells were significantly compromised by Erastin or RSL3 with a concentration range from 1 μM to 10 μM (Figure 1A–C). Additionally, the abilities of CAL33 and AMC-HN-8 cells were affected by Sorafenib at a concentration as low as 2 μM (Figure 1A–C), which also significantly impaired the survival of nasopharyngeal carcinoma cells (CNE-2 and S18) (Figure 1D,E). Only a high contraction of Sorafenib (more than 10 μM) affected the survival of the TU686 and S26 cells (Figure 1C,F). Surprisingly, nasopharyngeal carcinoma cells (CNE-2 and S18) were resistant to Erastin treatment, which slightly affected S26 at a concentration of 4 μM, as well as 10 μM (Figure 1D,E). Moreover, only a high level of RSL3 (10 μM for CNE-2 and S18; 4 μM for S26) reduced the viability of nasopharyngeal carcinoma cells (Figure 1D,E). To detect the toxicity of ferroptosis-inducers to non-cancer cells, mouse cortical neural stem cells (NSCs) were isolated and cultured from the embryonic stage of E14.5. Interestingly, none of the tested ferroptosis-inducers, including Erastin, RSL3, and Sorafenib, affected the neural stem cells’ viability (Figure 1G). On the other hand, all these drugs significantly reduced the viability of mouse neuroblastoma N2A cells, which have previously shown sensitivity to ferroptosis induction [21] even at the lowest concentration (1 μM) (Figure 1H). All these results suggest that different HNSCC cell lines show different responses to ferroptosis-inducers.

### 2.2. Ferroptosis-Inducer Treatment Affects Neither Proliferation nor Apoptosis in HNSCC Cells

To further investigate how ferroptosis-inducers affect HNSCC cell viability, we focused on studying the proliferation after Erastin or RSL3 treatment. HNSCC cells were incubated with EdU for 1 h after being treated with 10 μM of Erastin or RSL3. After fixation, the cells were stained with EdU and DAPI before being photographed under a fluorescence microscope, and the proportion of cells in the proliferative phase (EdU^+^/Hoechst^+^) was then calculated. Although cell viability was obviously affected, the proliferation portion of most tested cell lines, except AMC-HN-8, was not decreased after drug treatment (Figure 2A–F). To exam whether apoptosis cell death contributes to the survivability of HNSCC cells treated with Erastin or RSL3, the cleaved caspase-3 (CC3) level was examined. Western blotting analysis showed that none of the HNSCC cells expressed CC3 protein after being treated with the ferroptosis-inducers Erastin or RSL3 (Figure 2G) as compared to N2A cells that were treated with Staurosporine (STS), an inhibitor of protein kinase C (PKC) and an inducer of apoptosis. These data suggest that Erastin or RSL3 induced neither proliferation arrest nor apoptosis in HNSCCs.

### 2.3. Ferroptosis-Inducer Treatments Cause Different Ferroptosis Process in HNSCC Cells

To investigate whether ferroptosis cell death contributes to the distinct cell viability of HNSCCs, liproxstatin-1 (Lip-1), an inhibitor to ferroptosis, was applied to HNSCC cells before adding ferroptosis-inducers. A CCK8 assay showed that Lip-1 could significantly improve the cell viability affected by Erastin or RSL3 in tongue squamous cell carcinoma (CAL33) and laryngeal squamous cell carcinoma (AMC-HN-8 and TU686) (Figure 3A–C). However, Lip-1 could not restore the viability of the tested cell lines after a Sorafenib treatment nor the nasopharyngeal carcinoma cells (CNE-2, S18, and S26) after treatment with Erastin or RSL3 (Figure 3A–F). These data indicate that Erastin and RSL3 induce ferroptosis only in tongue squamous cell carcinoma cells and laryngeal squamous cell carcinoma cells but not in nasopharyngeal carcinoma cells, and the death of these HNSCC cells induced by Sorafenib was most likely not due to ferroptosis.

### 2.4. Ferroptosis-Inhibitor Reduces Cell Death and ROS Level in RSL3-Sensitive HNSCC Cells

To further exam whether HNSCC cells have different ferroptosis processes upon inducer treatment, cells were stained with propidium iodide (PI) for FACS analysis 12 h after being treated with 5 μM of RSL3 (Figure 4A,B). The population of PI-positive (PI^+^) cells was obviously increased following RSL3 treatment in the CAL33 and AMC-HN-8 cells (Figure 4A,B), which could be significantly reversed by ferroptosis-inhibitor Lip-1 (Figure 4A,B). Surprisingly, 5 μM of RSL3 did not change the portion of PI^+^ population in nasopharyngeal carcinoma cells CNE-2 (Figure 4C), suggesting a missing death in CNE-2 cells upon RSL3.

Ferroptosis in cells is accompanied by the upregulation of lipid peroxide from the cell membrane. The accumulation of these reactive oxygen species (ROS) will destroy the redox balance of cells, resulting in ferroptosis. To confirm there is a ferroptosis process in HNSCC cells, the ROS level was measured by using lipid-soluble small molecule BODIPY581/591 C11. A FACS analysis reveals that, similar to PI staining, RSL3 triggered an induction of ROS, which can be eliminated by Lip-1 only in tongue squamous (CAL33) and laryngeal squamous (AMC-HN-8) cell carcinoma (Figure 4D,E) but not in nasopharyngeal carcinoma cells (CNE-2) (Figure 4F). All these results suggest that, whereas tongue squamous and laryngeal squamous cell carcinoma are sensitive to ferroptosis-inducers, nasopharyngeal carcinoma cells are resistant to the ferroptosis induction.

### 2.5. Expression of GPX4 May Not Contribute to the Ferroptosis Process in HNSCC Cells

Glutathione peroxidase 4 (GPX4), an antioxidative enzyme and hallmark of ferroptosis [6], is generally downregulated upon ferroptosis [6]. To further confirm a ferroptosis process in HNSCCs, we measured the levels of GPX4 and heme oxygenase-1 (HO-1) in HNSCC cells treated with the ferroptosis-inducers Erasin or RSL3 (Figure 5A–C). The Western blot analysis data show an apparent reduction in the GPX4 level after being treated with 5 μM of Erastin or RSL3 in all tested HNSCC cell lines (CAL33, AMC-HN-8, and CNE-2), where 1 μM of RSL3 but not Erastin decreased GPX4 in CAL33 and CNE-2 cells (Figure 5A–C). Correspondingly, 5 μM of RSL3 upregulated the level of HO-1 firmly among these cells, and 1 μM of RSL3 only caused a high level of GPX4 in CNE-2 cells (Figure 5A–C). Moreover, either downregulated GPX4 or upregulated HO-1 trigged by 5 μM of RSL3 can be reversed by preincubating cells with the ferroptosis-inhibitor Lip-1 (Figure 5D–F). These results suggest that the expression of GPX4 or HO-1 is not responsible for the different viability outcomes after treatment with ferroptosis-inducers.

To further confirm the contribution of GPX4 or HO-1 to the different reactions of HSSCCs to ferroptosis-inducers, we compared the expression of GPX4 and HO-1 within all tested cells in our study. We found that tongue squamous cell carcinoma CAL33 expresses the highest level of GPX4 and the lowest level of HO-1 compared with other cell lines (Figure 5G–I). Moreover, laryngeal squamous cell carcinoma cells (AMC-HN-8 and TU686) expressed a moderate-high level of GPX4, and its level was low in all three nasopharyngeal carcinoma cells (CNE-2, S18, and S26) (Figure 5G–I). However, the nasopharyngeal carcinoma cells S18 expressed a relatively high level of HO-1 compared with CNE-2, S26, AMC-HN-8, and TU686 (Figure 5I), but this cell line presents a similar cell survival condition to CNE-2 and S26. Since a low level of GPX4 is generally associated with ferroptosis, our data suggest that the level of GPX4 or HO-1 in tested cells may not contribute to the sensitivity of HNSCCS to ferroptosis induction.

### 2.6. A Low Expression Level but Not Mutations of RAS Genes Correlates with the Sensitivity of HNSCCs to Ferroptosis-Inducers

It is well known that cells harboring oncogenic mutations of the *RAS* family (*KRAS*, *HRAS*, and *NRAS*) are susceptible to ferroptosis-inducers [11]. Therefore, we examined whether the HNSCC cells tested in this study contained any *RAS* mutations. mRNA was extracted from cells and then reverse-transcribed into cDNA for PCR amplification with primers to *RAS* family genes (Appendix A). The sequencing analysis data show that, although the C-terminus of *HRAS* shows a non-conserved region, as compared to the sequence of *KRAS* from the NCBI database, no other mutations or differences were found in *RAS* genes between the tested HNSCC cells (CAL33, AMC-HN-8, and CNE-2) and the NCBI templates (Appendix A). However, the qPCR analysis revealed that the mRNA expression of the *KRAS* gene in all ferroptosis-resistant nasopharyngeal carcinoma cells (CNE-2, S18, and S26) was much lower compared with tongue squamous cell carcinoma CAL33 and laryngeal squamous cell carcinoma (AMC-HN-8 and TU686) (Figure 6A). Additionally, the mRNA level of *HRAS* was significantly lower in nasopharyngeal carcinoma cells compared to CAL33 and AMC-HN-8 (Figure 6B). Although the mRNA level of *NRAS* was also dramatically low in CNE-2 cells compared with CAL33, it was expressed at a similar level among S26, AMC-HN-8, and TU686 cells (Figure 6C). These data suggest that a low mRNA level of *KRAS* or *HRAS* may be responsible for the resistance of nasopharyngeal carcinoma cells to ferroptosis-inducers.

### 2.7. The Expression of Ferroptosis-Related Genes (FRGs) in HNSCC Tissues and Cells

To further explore the molecular mechanism responsible for the sensitivity of HNSCCs to ferroptosis-inducers, we analyzed the expression of ferroptosis-related genes (FRGs) [30] in HNSCC patients. For that, the gene expression profile of 559 HNSCC patients and 136 control cohorts from the GEO database were enrolled, and the different expression FRGs were analyzed. We found that 91 FRGs (out of 256) were differentially expressed, with 21 being downregulated and 70 being upregulated (Figure 7A). KEGG pathway analysis showed that the top enrichment for the upregulated genes was the ferroptosis pathways, and 2-monocarboxylic acid metabolism was the top downregulated pathway enrichment (Figure 7A–C) in HNSCCs. Among these different expressed FRGs, whereas *ALOX12* (arachidonate 12-lipoxygenase, 12S type) and *TF* (transferrin) were strongly downregulated, *AURKA* (aurora kinase A), *TFRC* (transferrin receptor), and *FTH1* (ferritin heavy chain 1) were strongly upregulated. These findings indicate that these FRGs might be the potential molecular mechanisms involved in regulating ferroptosis in HNSCC.

To validate and explore whether the different expressed FRGs from HNSCC tissues are responsible for the sensitivity to ferroptosis-inducers, we tested the expression of these genes in HNSCC cell lines in this study using qPCR. We found that tongue squamous cell carcinoma and nasopharyngeal carcinoma cells (including CAL33, CNE-2, S18, and S26) have a similar level of both transferrin (*TF*) and transferrin receptor (*TFRC*) genes. The laryngeal squamous cell carcinoma cells (AMC-HN-8 and TU686) expressed a higher level of these genes than the other two HNSCC cell types (Figure 7D,E). Moreover, although the expression of the *AURKA* gene was high in CAL33 cells and low in S26 cells, it was expressed at the same level between S18 and TU686, relatively low in CNE-2 compared to S18, AMC-HN-8, and TU686 (Figure 7F). Surprisingly, the expression of ferritin heavy chain 1 (*FTH1*) was significantly higher in all nasopharyngeal carcinoma cells (CNE-2, S18, and S26) compared with tongue squamous cell carcinoma, as well as laryngeal squamous cell carcinoma cells (Figure 7G). The expression profile of the *FTH1* gene correlated well with the sensitivity pattern of these cells to ferroptosis-inducers (Figure 1, Figure 2, Figure 3 and Figure 4). Moreover, a high level of *FTH1* is associated with a poor prognostic effect on the overall survival of HNSCCs (Figure 7H). These data suggest that a high level of *FTH1* may contribute to the resistance of *KRAS*-low HNSCCs to ferroptosis induction.

### 2.8. Knockdown of FTH1 Sensitizes Ferroptosis-Numb Nasopharyngeal Carcinoma Cells

Our previous study showed that a shortage of *FTH1* sensitizing the neuroblastoma cells to ferroptosis-inducers [21]. We, therefore, hypothesize that a high level of this gene may contribute to the indolent property of HNSCC cells to RSL3 or Erastin treatment. To explore this, we first examined the expression of *FTH1* in CAL33 and its orthologous gene *Fth1* in N2A and found that *FTH1* in CAL33 was more than thirty times higher than *Fth1* in N2A (Figure 8A). FTH1 prevents hyper lipid oxidation and ferroptosis by storing free iron. Interestingly, whereas N2A cells harbor the lowest level of calcein–acetoxymethyl ester fluorescence, which can be loaded into living cells and quenched by binding it to labile iron, CNE-2 cells have the highest calcein–acetoxymethyl ester density, as compared to CAL33 and AMC-HN-8 cells (Figure 8B). These data indicate that CNE-2 cells, correlative with a high *FTH1* mRNA expression (Figure 8G), contain a low level of free iron, which may cause insensitive cells to ferroptosis.

To confirm that *FTH1* is responsible for the sensitivity of HNSCC to ferroptosis, we constructed GFP-tagged shRNA expression vectors to knock down *FTH1* (Figure 8C). CAL33, AMC-HN-8, and CNE-2 cells were then transfected either with a control vector expressing shRNA to target *luciferase* (sh*Luci*) or with vectors expressing shRNA to target *FTH1* (Figure 8D–F). A FACS analysis shows that more than 50% of the cells were positive with GFP in CAL33 and around 20% in the CNE-2 cells (Figure 8D). Nevertheless, the knockdown of *FTH1* not only increased the PI^+^ population in CAL33 cells (Figure 8E), but also led to a significantly high level of PI^+^ in RSL3 treated group, compared with the control (DMSO treatment) in the CNE-2 cells (Figure 8F). All these results indicate that a high level of *FTH1* may be blamed for the resistance of HNSCCs with low *KRAS* to ferroptosis-inducers.

### 2.9. Ferroptosis-Inducers and EGFR Inhibitors Play Synergetic Roles in Resistant HNSCCs

*EGFR* mutation plays a vital role in the occurrence and development of tumors. Many solid tumors highly express EGFR and, therefore, have been used as a target for treating different tumors, especially for HNSCCs [11,12]. To investigate whether a combination of ferroptosis-inducers and EGFR inhibitors play a synergetic effect in HNSCCs, we first studied the expression of *EGFR* in six kinds of HNSCC cells in this study. Our results showed that *EGFR* expression was higher in CAL33, S18, and S26 than in CNE-2, AMC-HN-8, and TU686 (Figure 9A). Then, cells were treated with a low level (0.2 or 1 μM) of RSL3 together with the clinically used EGFR inhibitor Gefitinib or monoclonal antibody Cetuximab (Figure 9B,C). Correlated with the expression level of *EGFR*, whereas Gefitinib or Cetuximab treatment significantly reduced the viability of CAL33 cells (Figure 9B), CNE-2 cells were entirely resistant to both Gefitinib or Cetuximab treatments (Figure 9C). Although the combination of RSL3 with Gefitinib did not change the viability of CAL33 cells compared to the Gefitinib-treated ones, RSL3 affected the survival of Cetuximab-treated cells (Figure 9B). Excitingly, RSL3 significantly reduced the viability of Cetuximab- and ferroptosis-inducer-resistant CNE-2 cells (Figure 9C). These data indicate that a combination of RSL3 and EGFR antibodies play a synthetic inhibiting effect in RSL3- and Cetuximab-resistant HNSCC cells.

## 3. Discussion

Head and neck squamous cell carcinoma (HNSCC) arises from the malignant transformation of the epithelial cells of the upper aerodigestive tract, constituting a heterogeneous group of tumors. Drugs with high efficiency, high selectivity, low toxicity, and reverse resistance to radiotherapy and chemotherapy are badly needed. Ferroptosis is a distinct form of regulated cell death that is iron-dependent and characterized by the accumulation of intracellular reactive oxygen species (ROS). It has gradually gained importance as an alternative to apoptosis for eliminating cancer cells and the regression of solid tumors [31], including HNSCCs [32]. We found that ferroptosis-inducers, as low as 1 μM of Erastin or RSL3, impair cell survival, induce PI^+^ cells, and elevate lipid ROS levels in tongue squamous carcinoma cells (CAL33) and nasopharyngeal carcinoma cells (AMC-HN-8 and TU686), all of which can be reversed by ferroptosis-inhibitor Lip-1 (Figure 1, Figure 3 and Figure 4). However, 4~5 μM of RSL3 neither affects cell viability nor induces PI^+^ cells or lipid ROS levels in nasopharyngeal carcinoma cells (CNE-2 and S18), which only respond to a high concentration (10 μM) of RSL3 and are entirely numb to Erastin (Figure 1, Figure 3 and Figure 4). Therefore, we conclude that sundry types of HNSCC cells have different sensitivities to ferroptosis-inducers.

GPX4 has a central regulator in ferroptosis by catalyzing the reduction in lipid peroxides. The inhibition of GPX4 activity directly by RSL4 or indirectly by Erastin by blocking the synthesis of GSH can lead to ferroptosis [33]. Therefore, a reduction in GPX4 is a standard marker to indicate ferroptosis. Interestingly, RSL3 and Erastin cause a decrease in GPX4 in nasopharyngeal carcinoma cells (CNE-2) without affecting cell viability, ROS levels, and cell death (Figure 1, Figure 3, Figure 4 and Figure 5). Moreover, the base level of GPX4 in nasopharyngeal carcinoma cells (CNE-2, S18, and S26) is much lower than CAL33 (Figure 5G). In the base situation, CNE-2 expresses a much lower level of *GPX4* compared to CAL33. The level of GPX4 decreased after treating cells with Erastin or RSL3 in CAL33 and AMC-HN-8 cells (Figure 5A–G). Consistent with the expression level of GPX4 in CNE-2, these cells contained relatively high lipid ROS compared with CAL33 and AMC-HN-8 cells (Figure 4D–F, left, marked by a dashed line). Given the fact that Lip-1 could not reverse cell viability defects caused by the high level of RSL3 (Figure 3D–F), nasopharyngeal carcinoma cells are insensitive to substances that can induce ferroptosis. So far, it is unclear how a high level of RSL3 affects cell viability, at least independent of apoptosis, since apoptosis marker cleaved caspase-3 is undetectable after RSL3 or Erastin treatment (Figure 2G).

Although nasopharyngeal carcinoma cells are resistant to RSL3 or Erastin, these cells are more sensitive to Sorafenib than tongue squamous carcinoma cells (CAL33) and nasopharyngeal carcinoma cells (AMC-HN-8 and TU686) (Figure 1 and Figure 3). Sorafenib, a protein kinase inhibitor with activity against many protein kinases, including VEGFR, PDGFR, and RAF kinases, is approved for treating primary kidney and advanced primary liver cancer [34]. Recently, it has been reported that Sorafenib is able to induce ferroptosis by inhibiting the cystine/glutamate antiporter system Xc- [35]. However, ferroptosis-inhibitor Lip-1 did not play any role in reversing the Sorafenib effect in all the tested HNSCCs (Figure 3A–F). Since Sorafenib can also induce apoptosis in hepatocellular carcinoma, the cell viability defect in HNSCCs may be caused by Sorafenib-mediated apoptosis. In fact, the anticancer efficacy of Sorafenib in the treatment of HNSCCS is not well studied. Therefore, the function and anticancer efficiency of Sorafenib in HNSCCs are worth further investigation.

Erastin or RSL3 can selectively kill cells harboring oncogenic *RAS* mutations, which activates RAS activity through ferroptosis. Furthermore, the genetic or pharmacological inhibition of RAS reduces the anticancer activity of Eerastin and RSL3 [22]. We found that all the HNSCC cell lines tested in this study did not contain mutations that can lead to the activation of RAS (Appendix A). However, all three nasopharyngeal carcinoma cells, including CNE-2, S18, and S26, expressed a very low mRNA level of *KRAS* compared with tongue squamous carcinoma and nasopharyngeal carcinoma cells (Figure 6A–C). It has been shown that oncogenic RAS signaling can increase cellular iron content by modulating the expression of iron metabolic genes such as *TfR1*, *FTH1*, and *FTL1* [22]. Indeed, we found that CNE-2 cells, which are resistant to Erastin and RSL3, contain a low level of free iron (or a high level of calcein–acetoxymethyl ester density) compared with other HNSCC cells susceptible to RSL3 and Erastin (Figure 8B). Consistent with this, CNE-2 and the other two nasopharyngeal carcinoma cell lines (S18 and S26) expressed a high level of *FTH1* (Figure 7G). Interestingly, HNSCC tumors expressed a significantly higher level of *FTH1* compared with normal tissues (Figure 7A). The elevated expression of *FTH1* is an unfavorable prognostic factor for the survival of HNSCC patients (Figure 7H). Additionally, the expression of *FTH1* mRNA is nearly 30 times higher in CAL33 than in neuroblastoma N2A cells, which are very sensitive to ferroptosis-inducers (Figure 7A) [21]. The interruption of *FTH1* not only enhances cell death in CAL33, but also sensitizes CNE-2 to RSL3 treatment (Figure 8E,F). All these data indicate that FTH1 functions to store free iron and, therefore, prevents cell ferroptosis, even though it has a relatively high level of ROS. Moreover, *FTH1* could be a good marker or a target for prediction therapy strategies for HNSCCs using the ferroptosis pathway.

Drug resistance in chemotherapy is a significant dilemma in the field of antitumors. The EGFR monoclonal antibody Cetuximab is approved by the FDA as a radiation sensitizer, alone or in combination with chemotherapy, for treating patients with recurrent or metastatic disease [36]. It has been shown that the EGFR inhibitor Gefitinib effectively blocks the growth of HNSCCs [37]. We found that nasopharyngeal carcinoma cells (CNE-2) also express a lower level of *EGFR* compared with CAL33, S18, and S26 (Figure 9A). Consistent with this, whereas both Cetuximab and Gefitinib treatments impair the viability of CAL33 cells, CNE-2 cells are also resistant to the EGFR inhibitor Gefitinib and the EGFR monoclonal antibody Cetuximab (Figure 9B,C). Excitingly, combining Cetuximab with RSL3, even with as low as 0.2~1 μM, significantly impaired the viability of CNE-2 cells compared with Cetuximab alone (Figure 9C). This synergetic effect was also taken in RSL3-treated CAL33 cells (Figure 9B). It is puzzling that a combination of RSL3 and Gefitinib has an ignorable effect on both CAL33 or CNE-2 cells. Nevertheless, a recent study has also shown that a cotreatment with β-elemene—a natural product isolated from the Chinese herb Curcumae rhizome—and Cetuximab sensitizes *KRAS* mutant metastatic colorectal cancer cells by inducing ferroptosis [38]. Their synergistic effect, via a combinative treatment with RSL3 and Cetuximab on ferroptosis- and Cetuximab-resistant HNSCC cells is interesting, which will provide a prospective therapeutic strategy for treating EGFR-resistant HNSCCs.

In summary, our study reveals that different types of HNSCCs have different sensitivities to ferroptosis-inducers, which are dependent on the expression level of *KRAS* and *FTH1*. FTH1 reduces the susceptivity of HNSCCs to ferroptosis-inducers by storing liable iron and therefore prevented ferroptosis. Combine this with an RSL3 treatment reverses the resistance of HNSCCS to the EGFR antibody Cetuximab.

## 4. Materials and Methods

### 4.1. The Mice

C57BL/6 pregnant mice were maintained in specific pathogen-free animal facilities at Sun Yat-Sen University (SYSU), and experiments were conducted according to licenses issued by the SYSU Institutional Animal Care and Use Committee (SYSU IACUC).

### 4.2. Data Collection and Analysis

Microarray data from HNSCC patients and healthy subjects were collected from the Gene Expression Omnibus (GEO) database (https://www.ncbi.nlm.NIH.gov/geo/, accessed on 5 June 2021). In total, 259 ferroptosis-related genes (FRGs) were downloaded from the ferroptosis database (FerrDb; http://www.zhounan.org/ferrdb, accessed on 5 June 2021) [30]. The differentially expressed genes were screened by using the R software package after normalization and batch effect treatment. Combined with FerrDatabase, ferroptosis-related, differentially expressed genes were obtained and applied for KEGG enrichment analysis. The prognosis prediction was analyzed on the GEPIA web server (http://gepia.cancer-pku.cn/index.html, accessed on 10 June 2022).

### 4.3. Construction of shRNA Expression Vectors

The construction of shRNA expression vectors was carried out as previously described [39]. The targeting sequences against *FTH1* are: sh*FTH1*-1 (GGTACCCAGGTGTTGTCTTTG) and sh*FTH1*-2 (GGATGAATCAGAAATCTATCC), synthesized by Tsingke Biotechnology Co., Ltd. (Beijing, China). Firstly, we prepared the dsDNA as the insertor using the 5′-phosphorylation of DNA and annealing oligos (incubated in 37 °C for 1 h, transferred to a 100 °C water bath for 10 min, and then, finally, decreasing temperature to 4 °C slowly). The shRNA expression cassata sequence luciferasere (sh*Luci*) and *FTH1* (sh*FTH1*) were cloned into the Sac I and Hind III sites of the empty vector pEGFP-U6+1. After ligation, we transformed plasmids into *E. coli.* for large scale amplification. We confirmed the plasmid by sequencing with U6 primer; plasmid extraction was carried out using the Endotoxin-Free Plasmid Mini Extraction Kit (Tiangen Biotech, Beijing, China).

### 4.4. Cell Culture

CNE2, S18, and S26 were cultured in RPMI 1640 (Corning, New York, NY, USA) containing 1% penicillin/streptomycin and 5% fetal bovine serum (FBS, Gibco, Carlsbad, CA, USA) at 37 °C and 5% CO_2_, and they were passed every three days.

CAL33, AMC-HN-8, TU686, 293T, and N2A cells were cultured in Dulbecco’s Modified Eagle Medium (DMEM, Gibco, Carlsbad, CA, USA) containing 1% penicillin/streptomycin and 10% fetal bovine serum (FBS, Gibco, Carlsbad, CA, USA) at 37 °C and 5% CO_2_. CAL33, AMC-HN-8, and TU686 were passed every three days; 293T and N2A were passed every two days.

Primary neural stem cells (NSCs) were isolated from mouse embryos at 14.5 days and then plated into T25 using the wall of not coated surface t(T25 in vertical position) and cultured in the neural stem cell medium (DMEM containing 1% B27, 1% penicillin/streptomycin, 20 ng/mL bFGF, and 20 ng/mL EGF). Neural stem cells formed neurospheres and were subcultured every 3 days. Before drug treatment, the neurospheres were disintegrated into single-cell suspension and then seeded in a culture dish coated with 0.1 mg/mL poly-L-lysine for adherent growth.

### 4.5. RNA Isolation and PCR Analysis

For RNA extraction, 1 mL TRIzol reagent (15596026, Thermo Scientific, Waltham, MA, USA) was added to the cells in the 6-well plate, and 0.2 mL chloroform was added after oscillation. We collected the lysate and centrifuged it. The upper colorless aqueous phase was taken, and 0.5 mL isopropanol was added. After centrifugation, the precipitation was collected, and we added 1 mL of 75% ethanol for washing. The precipitation was dried and dissolved in RNA-free water. The purity of RNA was assessed by Nanodrop three times for each sample (ND-ONE-W, Thermo Scientific, USA). The OD260/OD280 of pure RNA were between 1.8 and 2.0. The average RNA concentration of each sample was recorded for subsequent calculations.

For quantitative PCR (qPCR) analysis, 1 μg of the above RNA was used to synthesize first-strand cDNA using the RevertAid First Strand cDNA Synthesis Kit (K1622, Thermo Scientific, Waltham, MA, USA), according to the manufacturer’s instructions. Finally, these cDNA were used to analyze gene expression using the SYBR Green Premix (AG11701, Accurate, Changsha, China) in qPCR reactions, each in triplicate, on a real-time PCR system (StepOnePlus; Applied Biosystems, Waltham, MA, USA). *GAPDH* was used as the reference gene. The relative gene expressions (fold change) were calculated using the 2^−∆∆Ct^ method [40] and normalized to CAL33 (for Figure 6, Figure 7D–F, Figure 8A and Figure 9A) or the sh*Luci*-treated 293T control (see Figure 8C). The primers used in this study are presented in Appendix A.

### 4.6. Cell Viability Measurement (CCK-8 Assay)

Erastin (Selleck, Houston, TX, USA), RSL3 (Selleck, Houston, TX, USA), Sorafenib (Selleck, Houston, TX, USA), liproxstatin-1 (MCE, Princeton, NJ, USA), Cetuximab (Selleck, Houston, TX, USA), Gefitinib (Selleck, Houston, TX, USA), and staurosporine (GlpBio, Montclair, CA, USA) were dissolved in DMSO and kept at 80 °C.

HNSCC cells (CAL33, CNE-2, S18, S26, AMC-HN-8, U686) and N2A cells or primary neural stem cells (NSCs) were seeded into a 96-well plate with 8000 cells/wells, and DMSO, Erastin, RSL3, or Sorafenib were added into the medium the next day. Ferroptosis-inhibitors (liproxstatin-1) were added 2 h before Erastin and RSL3 or staurosporine. After culturing for 6 or 22 h, 10 μL of CCK-8 reagent (GlpBio, Montclair, CA, USA) was added and incubated at 37 °C and 5% CO_2_ for 2 h. The absorbance was measured at 450 nm with a microplate analyzer.

### 4.7. EdU Labeling and Staining

Six kinds of HNSCC cells (CAL33, CNE-2, S18, S26, AMC-HN-8, TU686) were separately seeded on slides and cultured at 37 °C and 5% CO_2_. After drug treatment, EdU (final concentration: 10 μM; Sigma, St. Louis, MO, USA) was added to the medium and incubated for 1 h. The cells were fixed with 4% paraformaldehyde, treated with 2N HCl at 37 °C for 30 min, and sealed and made permeable with blocking solution (BS) (5% goat serum, 1% bovine serum albumin, 0.4% Triton X-100). BrdU antibody (Abcam, Cambridge, UK, 1:200 dilution) was incubated overnight at 4 °C, followed by secondary antibody Alexa Fluor 488 (Invitrogen, Carlsbad, CA, USA, 1:200) for 1 h at room temperature and DAPI for 20 min. A mounting medium (Sigma, St. Louis, MO, USA) was used to seal the plates. Finally, the images were observed under a fluorescence microscope (ZEISS, Oberkochen, Germany).

### 4.8. Propidium Iodide (PI) Staining

For flow cytometry detection, 5 mM propidium iodide (Beyotime, Shanghai, China) was directly added into the medium to a final concentration of 5 μM (except for the unstained sample for negative control) and incubated at 37 °C for 25 min. After washing with PBS (1×), HNSCC cells (CAL33, CNE-2, or AMC-HN-8) were digested into single cells with trypsin and then the percentage of PI-positive cells was analyzed by FCM. The detection channel was PE (PI-DNA excitation light at 535 nm and emission light at 617 nm). Moreover, cells (CAL33, CNE-2, AMC-HN-8) transfected by sh*Luciferase* were sorted with an GFP-positive gate in the FITC channel and then the percentage of PI-positive cells in the PE channel was analyzed with FCM. For fluorescence microscope observation, the medium containing PI and Hoechst was directly added and incubated at 37 °C for 25 min to a final concentration of 5 μM and 10 μg/mL, respectively, before being used for microscope imaging.

### 4.9. Lipid ROS Detection

On the day before drug treatment, 2.5 × 10^5^ cells (CAL33, CNE-2, S18, S26, AMC-HN-8, TU686) were plated to each well of a 6-well plate with 2 mL of medium per well. A control well (plated cells without treatments) was required. After a ferroptosis-inducer treatment for 24 h, BODIPY 581/591 C11 (Thermo Fisher, Waltham, MA, USA) dissolved in DMSO was added to the medium at a concentration of 2 mM. Then, the cells were cultured at 37 °C and 5% CO_2_ for 20 min. After staining, the cells needed to be digested into a single cell with trypsin. Washing with PBS twice, the cells were collected into a 2 mL Eppendorf tube and were ready to be analyzed with FCM (CytoFLEX, Beckman Coulter, Brea, CA, USA). The detection channel was FITC (BODIPY 581/591C11 excitation light at 488 nm and emission light at 530 nm).

### 4.10. Measurement of Labile Iron Pool (LIP)

HNSCC cells (CAL33, CNE-2, AMC-HN-8) or N2A cells at a density of 1 × 10^6^/mL were stained with 0.6 μM of calcein–acetoxymethyl ester (Beyotime, Shanghai, China), a fluorescence probe, for 30 min at 37 °C and 5% CO_2_. Then, the cells were washed twice with PBS (1×) and either incubated with 100 μM deferiprone (DFO) for 1 h at 37 °C or left untreated. Finally, the cells were collected into a 2 mL Eppendorf tube and were ready to be analyzed with FCM (CytoFLEX, Beckman Coulter, USA) with the FITC channel (calcein excitation light at 494 nm and emission light at 514 nm). The difference in the mean cellular fluorescence with and without DFO incubation reflected the amount of LIP. The peak plot was formed using the Flowjo software (X.0.7) (BD, Ashland, KY, USA).

### 4.11. Western Blotting

After a drug treatment using DMSO, Erastin, RSL3, Sorafenib, or in combination with liproxstatin-1, the cells (CAL33, CNE-2, S18, S26, AMC-HN-8, TU686) were lysed with NETN buffer (50 mM Tris-HCl pH 7.4, 150 mM NaCl, 1% NP40, 1mM EDTA, plus one tablet of Roche complete protease inhibitor per 10 mL) to collect the total protein of the cells. After SDS-PAGE, the protein on the gel was transferred to a PVDF membrane (Bio-Rad, Hercules, CA, USA) and sealed with 5% milk for 1 h. Then, the primary antibody GPX4 (ABclonal, 1:1000 dilution), HO-1 (Proteintech, Chicago, IL, USA, 1:1000), GAPDH (ABclonal, 1:5000), beta-actin (ABclonal, Wuhan, China, 1:5000), or cleaved caspase-3 (Cell Signaling Technology, Boston, MA, USA, 1:1000) were incubated overnight at 4 °C. The secondary antibody with HRP was incubated at room temperature for 1 h. Proteins were visualized with the SuperSignal Chemiluminescent Substrate (Thermo Scientific, Waltham, MA, USA).

### 4.12. Statistical Analysis

The data are shown as mean ± SEM. Statistical significance (*p*-values) was calculated by Prism 8.0.2 (GraphPad Software, San Diego, CA, USA). The two-tailed *t*-test, a one-way ANOVA test, and a two-way ANOVA test were used in the present study. More details of statistical analysis are described in each figure’s legend.

## 5. Conclusions

In summary, our study found that, attributed to the different expressions of *KRAS* and ferritin heavy chain (*FTH1*), HNSCC cells respond differently to ferroptosis induction. A high level of *FTH1* is associated with the poor prognostic survival of HNSCCs, and the knockdown of *FTH1* promotes HNSCC cell death triggered by the ferroptosis-inducer RSL3. Additionally, nasopharyngeal carcinoma cells (CNE-2), which express a high level of *FTH1* and a low level of *EGFR*, are insensitive to ferroptosis induction and EGFR inhibition. However, the ferroptosis-inducer RSL3 and the EGFR monoclonal antibody Cetuximab play a synthetic role in inhibiting the survival of CNE2 cells. Our findings show that *FTH1* is a vital player in ferroptosis resistance in HNSCCs and that a combination of ferroptosis-inducers and EGFR inhibitions could be a promising target strategy for HNSCC therapy.

## Figures and Tables

**Figure 1 ijms-23-09014-f001:**
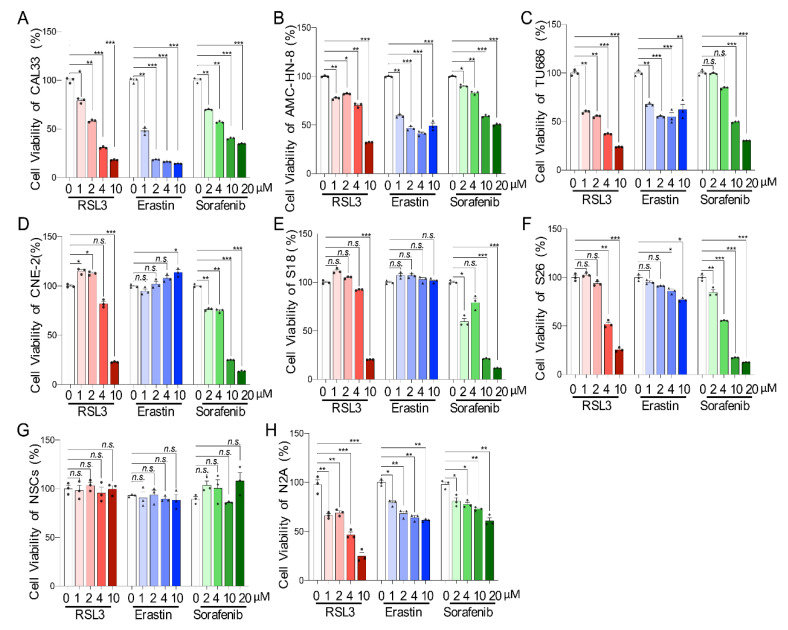
HNSCC cells are sensitive to ferroptosis-inducers. (**A**–**F**) The cell viability of CAL33 (**A**), AMC-HN-8 (**B**), TU686 (**C**), CNE-2 (**D**), S18 (**E**), and S26 (**F**) were tested by CCK-8 24 h after treatment with the indicated concentrations of three types of ferroptosis-inducers (RSL3, Erastin, or Sorafenib). (**G**,**H**) Cell viability of neural stem cells (NSCs) (**G**) and N2A cells (**H**) tested by the same method as the HNSCC cells as the control group. *n* = 3. The mean ± SEM of three independent experiments is shown. A two-way ANOVA, followed by an uncorrected Fisher’s LSD test, was used for statistical analysis. * *p* < 0.05; ** *p* < 0.01; *** *p* < 0.001; *n.s.*: not significant. Black triangles: number of experiments with RSL3 treatment; dots: number of experiments with Erastin treatment samples; diamonds: number of experiments treated with Sorafeninb.

**Figure 2 ijms-23-09014-f002:**
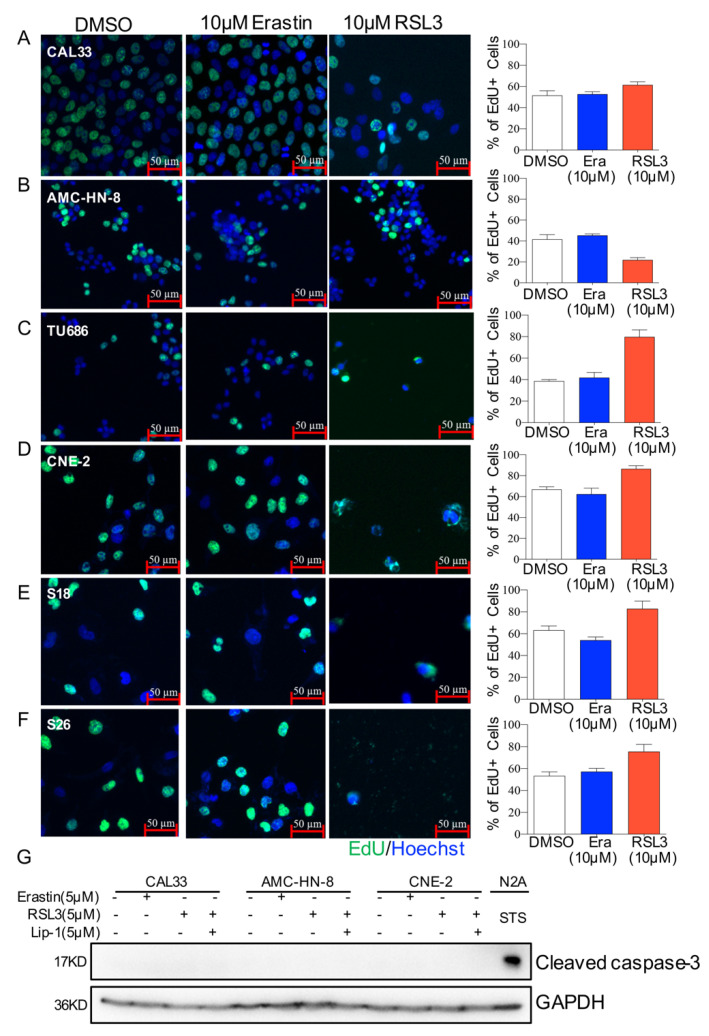
Morphology of HNSCC cells after ferroptosis-inducer treatment. (**A**–**F**). Representative images of EdU staining in CAL33 (**A**), AMC-HN-8 (**B**), TU686 (**C**), CNE-2 (**D**), S18 (**E**), and S26 (**F**). Cells were labeled for 1 h after treating for 23 h with DMSO, 10 μM of Erastin or RSL3. The cells were then fixed for EdU antibody staining. Images represent two independent experiments. Scale bar = 50 μm. The quantification of the average EdU-positive cells (%) of each cell line is shown on the right. Era: Erastin. (**G**) HNSCC cells were treated with DMSO, 5 μM Erastin, 5 μM RSL3, or 200 nM STS for 24 h, with or without 5 μM liproxstatin-1 treated for 2 h in advance. Protein extraction was prepared for Western blotting analysis with antibodies against cleaved caspase-3 and GAPDH. The images represent three independent experiments.

**Figure 3 ijms-23-09014-f003:**
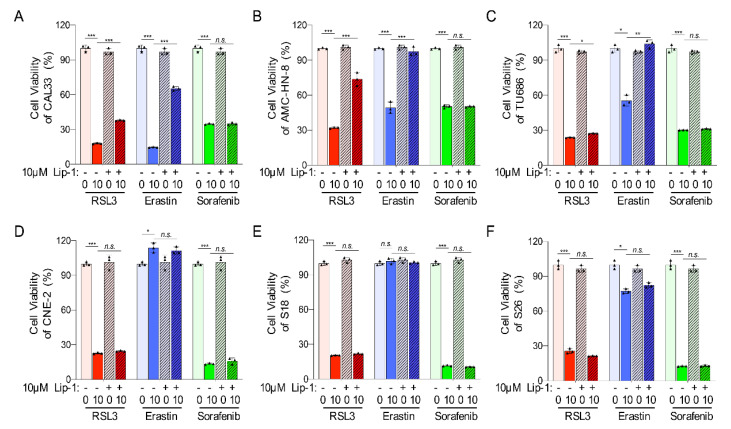
Ferroptosis inhibitor liproxstatin-1 can block ferroptosis in HNSCC cells. (**A**–**F**) The viability of CAL33 (**A**), AMC-HN-8 (**B**), TU686 (**C**), CNE-2 (**D**), S18 (**E**), and S26 (**F**) was tested by CCK-8 at 24 h after treatment with ferroptosis-inducers (RSL3, Erastin, or Sorafenib), with or without a 10 μM liproxstatin-1 for 2 h in advance. The mean ± SEM of three independent experiments is shown. A two-way ANOVA, followed by an uncorrected Fisher’s LSD test, was used for statistical analysis. * *p* < 0.05; ** *p* < 0.01; *** *p* < 0.001; *n.s.*: not significant. Black triangles: number of experiments.

**Figure 4 ijms-23-09014-f004:**
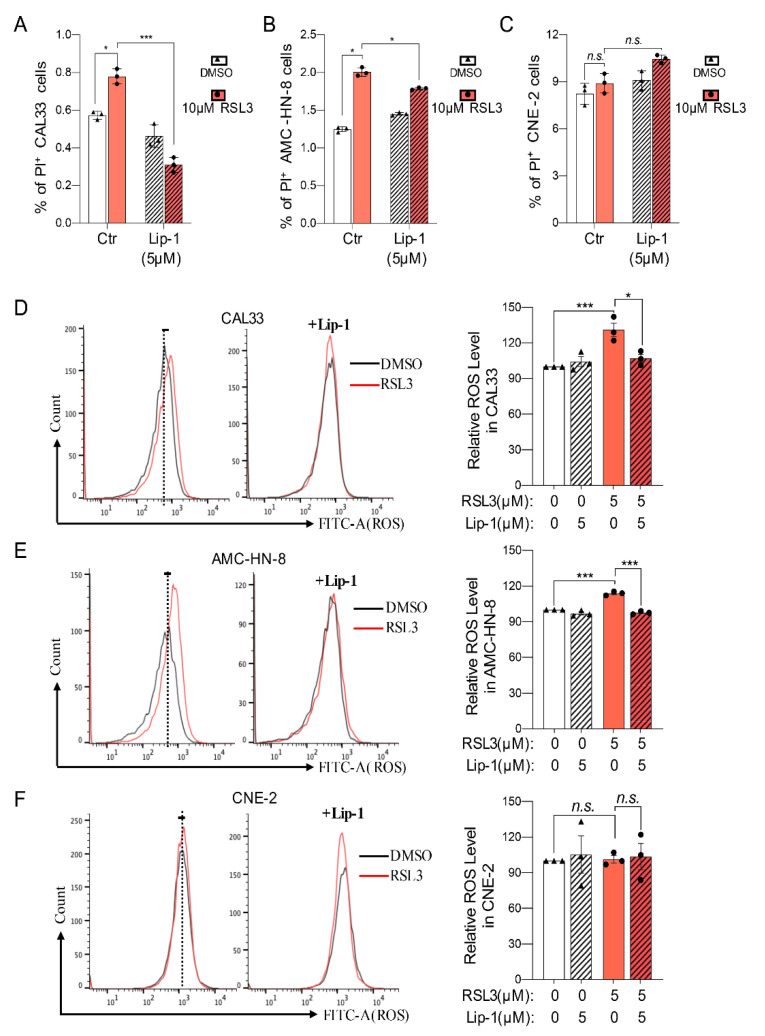
HNSCC cells are sensitive to ferroptosis-inducers. (**A**–**C**) The percentage of PI-positive cells of CAL33 (**A**), CNE-2 (**B**), and AMC-HN-8 (**C**) at 24 h after treating with 5 μM RSL3, with or without 10 μM liproxstatin-1 treated for 2 h in advance. The mean ± SEM of three independent experiments is shown. A two-way ANOVA, followed by an uncorrected Fisher’s LSD test, was used for statistical analysis. (**D**–**F**) ROS level in CAL33 (**D**), CNE-2 (**E**), and AMC-HN-8 (**F**) after ferroptosis-inducer (5 μM RSL3) treatment, with or without 10 μM liproxstatin-1 treated for 2 h in advance. Dot lines high line the mean of ROS level of the DMSO controls. Gray lines for DMSO treatment. Red lines for 5 μM RSL3 treatment. The histogram on the right shows the absolute fluorescence value. Black triangles represent the number of DMSO-treated experiments. Black dots represent the number of experiments with 5 μM RSL3-treated. The mean ± SEM of three independent experiments is shown. A two-way ANOVA, followed by an uncorrected Fisher’s LSD test, was used for statistical analysis. * *p* < 0.05; *** *p* < 0.001; *n.s.*: not significant.

**Figure 5 ijms-23-09014-f005:**
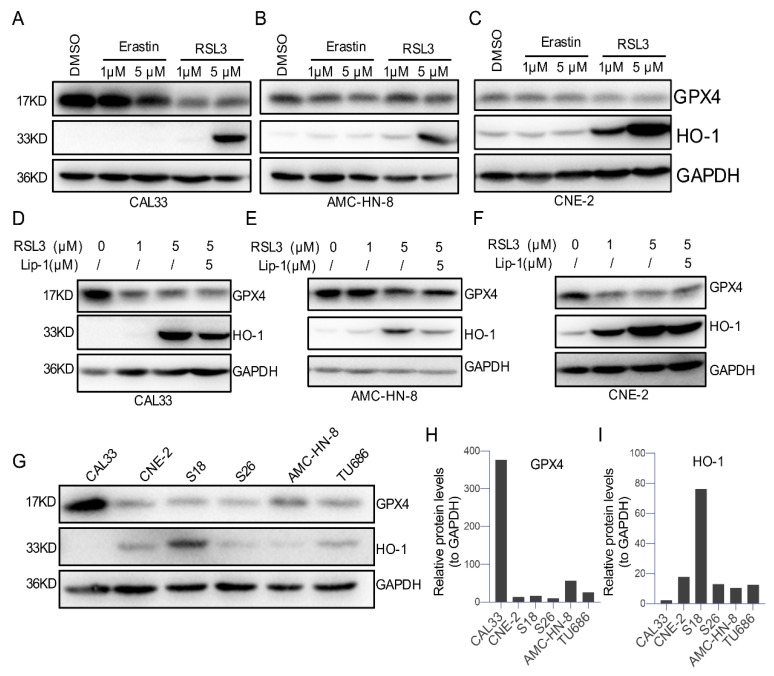
GPX4 and HO-1 were differently expressed after ferroptosis-inducer treatment in different HNSCC cells. (**A**–**C**) Representative HNSCC cells were treated with DMSO, Erastin, or RSL3 for 24 h. Western blotting analysis conducted with antibodies against HO-1, GPX4, and GAPDH. (**D**–**F**) Representative HNSCC cells were treated with DMSO or RSL3 for 24 h, with or without 5 μM liproxstatin-1 treated for 2 h in advance. Western blotting analysis was carried out, as mentioned above. (**G**) The background protein level of GPX4 and HO-1 in different HNSCC cells. Representative immunoblots of three independent experiments are shown above. (**H**,**I**) The quantification of (**G**).

**Figure 6 ijms-23-09014-f006:**
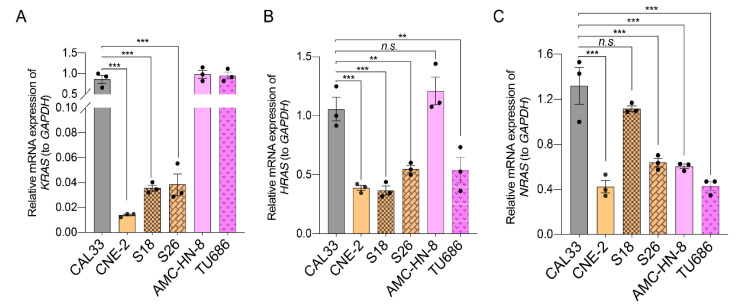
Expression of RAS family genes in HNSCC cells. (**A**–**C**) A quantification polymerase chain reaction (qPCR) was used to quantify the expression of *KRAS* (**A**), *HRAS* (**B**), and *NRAS* (**C**). The reference genes used were *GAPDH*. The mean ± SEM of three independent experiments are shown. A one-way ANOVA, followed by a Dunnett’s multiple comparisons test, was used for statistical analysis. ** *p* < 0.01; *** *p* < 0.001; *n.s.*: not significant. Black dots represent number of experiments.

**Figure 7 ijms-23-09014-f007:**
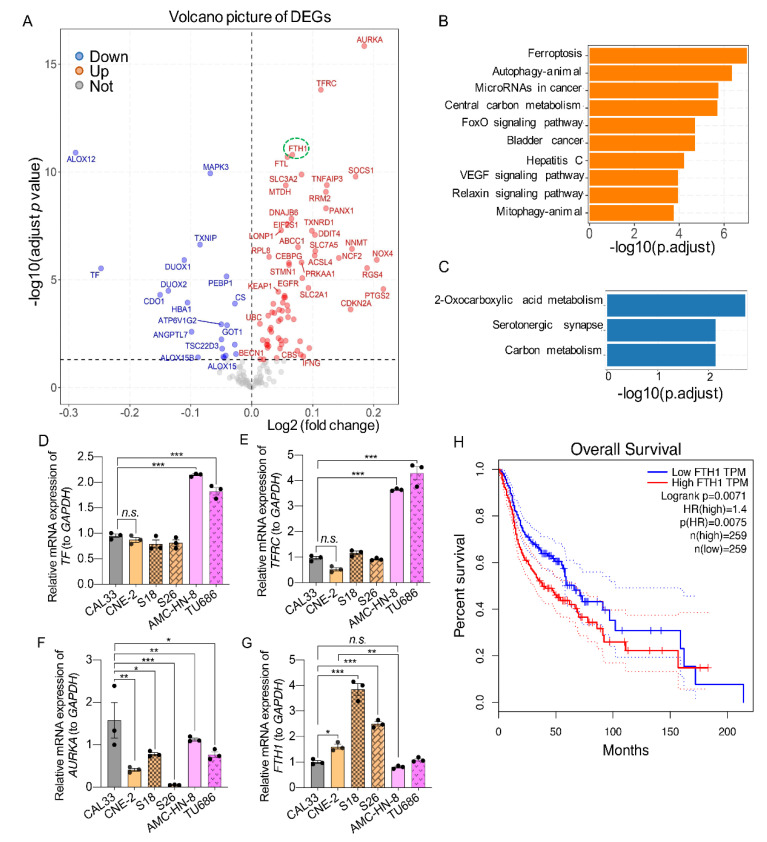
Identification of candidate ferroptosis-related genes in HNSCCs tissues. (**A**) A volcano map of differently expressed genes (DEGs) related to ferroptosis in the GEO database. The X-axis represents the fold change of the difference after conversion to log2, and the Y-axis represents the significance value after conversion to −log10. Red represents upregulated DEGs, blue represents downregulated DEGs, and gray represents non-DEGs. The false discovery rate (FDR < 0.05) related to a different expression. (**B**) The GO biological process enrichment analysis of the upregulated DEGs. (**C**) The three most significant KEGG pathways in GEO cohorts. The X-axis is the number of genes annotated to a category of KEGG pathway, and the Y-axis is the category of the KEGG pathway. (**D**–**G**) The relative expression of indicated ferroptosis-related genes (FRGs) in HNSCC cells. The reference genes used were *GAPDH*. The mean ± SEM of three independent experiments is shown; *n* = 3. A one-way ANOVA, followed by a Dunnett’s multiple comparisons test, was used for statistical analysis. * *p* < 0.05; ** *p* < 0.01; *** *p* < 0.001; *n.s.*: not significant. Black dots represent number of experiments. (**H**) Kaplan-Meier survival curves of the prognostic signature of FTH1 in HNSCCs. Blue line represent the over all survival of FTH1 level group; Red line represent the over all survival of high FTH1 group TPM. 95% confidence interval is presented in with dash lines.

**Figure 8 ijms-23-09014-f008:**
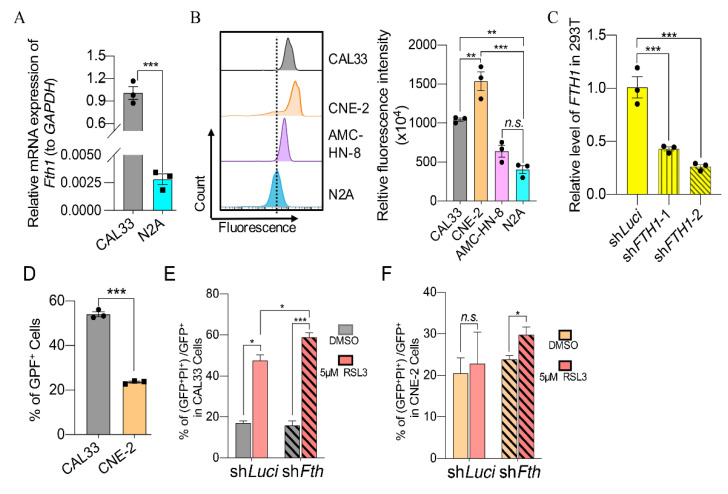
*FTH1* is responsible for the sensitivity of HNSCC cells to ferroptosis. (**A**) The relative mRNA level of *FTH1* (to *GAPDH*) between CAL33 and N2A. The mean ± SEM of three independent experiments are shown. The *p*-values are from an unpaired Student’s *t*-test. (**B**) FACS-detected labile iron pool in the representative HNSCCs and N2A cells. The X-axis represents the fluorescence of calcein–acetoxymethyl ester. The Y-axis represents the count of cells. Cells were incubated with a 0.6 μM calcein–acetoxymethyl ester fluorescence probe for 1 h before being analyzed with FACS. The histogram shows the quantification of relative fluorescence intensity. The mean ± SEM of three independent experiments is shown. A one-way ANOVA, followed by a Dunnett’s multiple comparisons test, was used for statistical analysis. (**C**) The relative expression of *FTH1* after targeting by shRNA in 293T cells. *n* = 3. The *p*-values are from a two-tailed *t*-test. (**D**) The percentage of GFP^+^ cells in CAL33 and CNE-2 48 h after transfection with GFP-tagged shRNA expression vector. *n* = 3. A one-way ANOVA, followed by a Dunnett’s multiple comparisons test, was used for statistical analysis. (**E**,**F**) The percentage of PI-positive cells in GFP-positive CAL33 (**E**) and CNE-2 (**F**) cells. HNSCC cells were transfected with sh*Luciferase* or sh*Fth* for 24 h and then treated with 5 μM RSL3 for another 24 h. Cells were stained by PI and collected for FCM detection. The data were obtained from three independent experiments. The mean ± SEM are indicated. A two-way ANOVA, followed by an uncorrected Fisher’s LSD test, was used for statistical analysis. * *p* < 0.05; ** *p* < 0.01; *** *p* < 0.001; *n.s.*: not significant. Black dots or boxes represent number of experiments.

**Figure 9 ijms-23-09014-f009:**
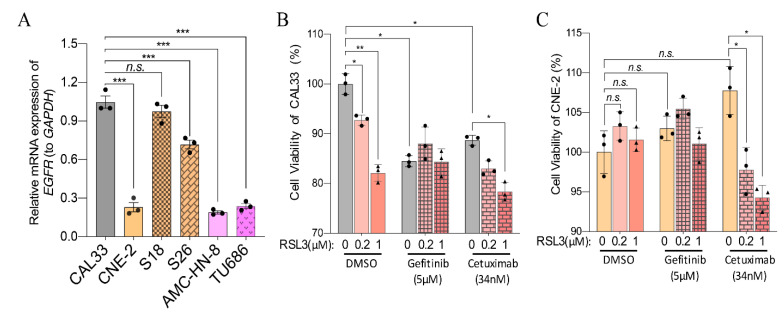
The combination of ferroptosis-inducers and EGFR inhibitors reduces the sensitivity to ferroptosis-inducers. (**A**) Relative expression of *EGFR* in different HNSCC cells. The reference genes used were *GAPDH*. The mean ± SEM of three independent experiments are shown3. A one-way ANOVA, followed by a Dunnett’s multiple comparisons test, was used for statistical analysis (*n.s.*: not significant; *** *p* < 0.001). (**B**,**C**) The viability of CAL33 (**B**) and CNE-2 (**C**) at 24 h after treatment with the indicated concentrations of ferroptosis-inducers and EGFR inhibitors. The data were obtained from three independent experiments. Mean ± SEM is indicated. A two-way ANOVA, followed by an uncorrected Fisher’s LSD test, was used for statistical analysis. * *p* < 0.05; ** *p* < 0.01; *** *p* < 0.001; *n.s.*: not significant. Black dots or triangles represent number of experiments.

## Data Availability

The data presented in this study are available upon request from the corresponding authors.

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
