# Peer review of "Combination RSL3 Treatment Sensitizes Ferroptosis- and EGFR-Inhibition-Resistant HNSCCs to Cetuximab"

_ijms, 2022, doi:10.3390/ijms23169014_

Round 1

Reviewer 1 Report

Interesting work, but a few things are not clear:

1) the methodology is described very briefly, for example, the section "RNA isolation and PCR analysis", it is not known how the purity of RNA was assessed, the amount of RNA was used for reverse transcription, how many repetitions were performed, what results were obtained and what reference gene was measured in which tissue the results were compared

2) this applies to all sections

3) the results of statistical analyzes are presented only in figures and charts; it is not clear what statistical tests were analyzed (especially on such small analyzed numbers)

Author Response

Interesting work, but a few things are not clear:

 Response 1:

We thank the Reviewer for her/his very positive evaluation of our manuscript.

1) the methodology is described very briefly, for example, the section "RNA isolation and PCR analysis", it is not known how the purity of RNA was assessed, the amount of RNA was used for reverse transcription, how many repetitions were performed, what results were obtained and what reference gene was measured in which tissue the results were compared.

 Response 2:

Many thanks to the Reviewer for her/his very constructive suggestions. Following her/his suggestions, we have now included more detailed information for the "RNA isolation and PCR analysis" section, and There information is described in lines 634~643 in the revised manuscript.

2) this applies to all sections

 Response 3:

We thank this Reviewer for her/his constructive comments. More detailed information for the methodology is now described and included. Please find this information in lines 574~577, lines 589~597, lines 621~627, lines 648~649, line 674, lines 685~692, lines 698~700, and line 702 in the revised manuscript.

3) the results of statistical analyzes are presented only in figures and charts; it is not clear what statistical tests were analyzed (especially on such small analyzed numbers)

 Response 4:

Many thanks to the Reviewer for her/his constructive suggestions. Following her/his suggestions, we have included all the statistical test methods in each figure legend. Please find this information in lines 147~148, lines 178~179, lines 208~210, lines 247~249, lines 252~253, lines 282~283, lines 233~234, lines 391~392, lines 423~424, lines 428~432, lines 435~436, and lines 464~468 in the revised manuscript.  

Reviewer 2 Report

The introduction is very useful to understand the scope of the paper.

The experiment reported in Figure 1 is missing of a negative control.

the significance of the data in figure are not quantified with accuracy. please update.

a non-cancerous cell line must be included to exclude toxicity.

figure 7 is of low quality. please substitute it.

i suggest to include the conclusion section in order to better clarify the data.

Author Response

The introduction is very useful to understand the scope of the paper.

 Response 5:

We thank the Reviewer for her/his very positive evaluation of our manuscript.

The experiment reported in Figure 1 is missing of negative control.

 Response 6:

Many thanks to the Reviewer for her/his constructive suggestions. Following her/his suggestions, we have now included primary neural stem cells (NSCs) as a negative control. These data are now included in Figure 1Gand described in lines 132~135 in the revised manuscript. In addition, we also included neuroblastoma N2A as a positive control, which is included in Figure 1H and described in lines 135~138 in the revised manuscript.

the significance of the data in figure are not quantified with accuracy. please update.

 Response 7:

We thank this viewer for her/his vital comment. Following her/his suggestions, we have now provided accurate information on the significance of the data in each figure legend and updated this information in all the figures(except figure 2 & 5)in the revised manuscript.

a non-cancerous cell line must be included to exclude toxicity.

 Response 8:

Many thanks to the Reviewer for her/his constructive suggestions and vital comment. Following her/his suggestions, we have included primary neural stem cells (NSCs) as a non-cancerous cell line to test the toxicity. Interestingly, all the ferroptosis inducers used in this study did not show any toxicity to NSCs with the tested concentration. These data are now included in Figure 1G and described in lines 132~135 in the revised manuscript.

figure 7 is of low quality. please substitute it.

 Response 9:

We thank this Reviewer for her/his vital comment. Following her/his suggestions, we have now replaced Fig 7 with enlarged and high-quality pictures in lines 376~381 in the revised manuscript.

i suggest to include the conclusion section in order to better clarify the data.

Response 10:

We thank this Reviewer for her/his constructive suggestions. Following her/his suggestions, we have now added a conclusion section which is described in lines 739~750 in the revised manuscript.

Reviewer 3 Report

The presentation of the results can be improved. Bar charts were mainly used, but scatter plots are now the standard. A uniform scaling of the y-axis would also be advantageous for a better recording of the results. The color scheme of the individual diagrams in the various figures is different and is also not obvious to the reader. 

Author Response

The presentation of the results can be improved. Bar charts were mainly used, but scatter plots are now the standard. A uniform scaling of the y-axis would also be advantageous for a better recording of the results. The color scheme of the individual diagrams in the various figures is different and is also not obvious to the reader

Response 11:

We thank this Reviewer for her/his positive evaluation of our manuscript and constructive suggestions. Following her/his suggestions, we have updated and replaced all figures (except figure 5) with bar charts with scatter plots, uniform scaling of the y-axis, and the color scheme of the individual diagrams in the revised manuscript.

Round 2

Reviewer 1 Report

I have read the revised manuscript and in my opinion it still does not meet IJMS publishing standards:

1) the entire manuscript does not use the correct spelling of the names of human genes and proteins, e.g. line 58 or 96, but not only. When writing about genes, the authors use abbreviations of the names of these genes in a way that corresponds to the nomenclature of proteins

2) I really don't know what the final sentence of the fourth introductory paragraph means: "Moreover, whether ferroptosis inducers play a synergetic function with other drugs, such as EGFR, in killing HNSCCs is unclear"

3) The description of statistical methods is still laconic. The authors added important information to the figure signatures, e.g. "Mean ± SEM of three independent experi248 ments is shown. P values from two-way ANOVA, NEJM. N.s. for p≥0.05, not significant; * for p <0.05; ** for p <0.01; *** for p <0.001. ", but it is still unclear how they calculated these significant or insignificant differences. The figures are derived from the analysis of qualitative or quantitative data (numbers) and this information is still missing from the manuscript. What "NEJM" means in that quted descriptions?

4) The description of the methodology still lacks important information, such as "For RNA extractions .... was aded to the cells .....", "Cells were seeded on slides and cultured" - what cells ??? (line 529,550, 557)

5) no raw data, no information on what was the control sample for the expression comparison, no information on how the difference in expression was calculated

6) All the details and information on the real-time PCR technique are contained in one sentence:"Finally, these cDNA 641 was used to analyzed gene expression by SYBR Green Premix (AG11701, Accurate, 642 Changsha, Hu-nan, China) in quantitative PCR (qPCR) reactions". There is even no information on which device (thermocycler) experiments were performed, what standards were used for the standard curve and no raw data or information where they were made available from these quantitative/numerical data

Author Response

To Review 1,

I have read the revised manuscript and in my opinion it still does not meet IJMS publishing standards:

1) the entire manuscript does not use the correct spelling of the names of human genes and proteins, e.g. line 58 or 96, but not only. When writing about genes, the authors use abbreviations of the names of these genes in a way that corresponds to the nomenclature of proteins

 Response 1:

Many thanks to the Reviewer for her/his very vital comments. Following her/his comments, we have corrected the spelling of the gene name: adding italic to all the names for the human gene, mRNA, or cDNA. The correction is applied in the main text and all the figures (Fig 6A-C, Fig 7D-G, Fig 8A, Fig 8C, and Fig 9A).

2) I really don't know what the final sentence of the fourth introductory paragraph means: "Moreover, whether ferroptosis inducers play a synergetic function with other drugs, such as EGFR, in killing HNSCCs is unclear".

 Response 2:

We thank this Reviewer for pointing out the issue and sincerely apologize for the mistake in the sentence mentioned above. We have now corrected the sentence: "Moreover, whether ferroptosis inducers play a synergetic function with other drugs, such as inhibitors or antibodies against EGFR, in killing HNSCCs is unclear.". Please find this correction in line 110 in the revised manuscript.

3) The description of statistical methods is still laconic. The authors added important information to the figure signatures, e.g. "Mean ± SEM of three independent experi248 ments is shown. P values from two-way ANOVA, NEJM. N.s. for p≥0.05, not significant; * for p <0.05; ** for p <0.01; *** for p <0.001. ", but it is still unclear how they calculated these significant or insignificant differences. The figures are derived from the analysis of qualitative or quantitative data (numbers) and this information is still missing from the manuscript. What "NEJM" means in that quted descriptions?

 Response 3:

We thank this Reviewer for her/his comments. We have now specified the detailed methods used for statistical calculation and included them in each figure legend in the revised manuscript. NEJM, the New England Journal of Medicine, stands for the new statistical reporting style announced by the NEJM. We have deleted this word in the revised manuscript not to confuse the readers.

4) The description of the methodology still lacks important information, such as "For RNA extractions .... was aded to the cells .....", "Cells were seeded on slides and cultured" - what cells ??? (line 529,550, 557)

 Response 4:

Many thanks to the Reviewer for her/his vital comments. Following her/his comments, we have specified the detailed cell line information in the "Materials and Methods” section. Please find this information in the revised manuscript on lines 683~684, lines 703~704, line 718, line 721, 727~728, line 738, and line 749.

5) no raw data, no information on what was the control sample for the expression comparison, no information on how the difference in expression was calculated

 Response 5:

We thank this Reviewer for her/his vital comments. The different expression is compared to CAL33 (for Fig 6, Fig7D-F, Fig 8A, Fig 9A) or shLuci-treated 293T control (for Fig 8C). We used the delta-delta Ct method to calculate the differential expression. Following her/his comments, we have included this information in the "Materials and Methods" section. Please find this information in lines 672~676 in the revised manuscript. Meanwhile, we have now offered the detailed data for all our statistical analyses as supplemental data and submitted them to the MDPI system.

6) All the details and information on the real-time PCR technique are contained in one sentence:"Finally, these cDNA 641 was used to analyzed gene expression by SYBR Green Premix (AG11701, Accurate, 642 Changsha, Hu-nan, China) in quantitative PCR (qPCR) reactions". There is even no information on which device (thermocycler) experiments were performed, what standards were used for the standard curve and no raw data or information where they were made available from these quantitative/numerical data

 Response 6:

We thank this Reviewer for her/his vital comments. We apologize for missing the details information regarding the real-time PCR technique. More information, including the device, was added to the "Materials and Methods" section. Since we use the PCR efficiency approach to assess the relative level of tested genes, we did not include any standard curve analysis after validating our PCR primers. Different expression is normalized to CAL33 (for Fig 6, Fig7D-F, Fig 8A, Fig 9A) or shLuci-treated 293T control (for Fig 8C). This information is now included in lines 672~676 in the revised manuscript. Meanwhile, we also offered the detailed data for all our statistical analyses as supplemental data and submitted them to the MDPI system (See also Response #5). More detailed information, such Ct value, can be provided upon requirement.

Round 3

Reviewer 1 Report

My recommendation: accept with minor revision.

The manuscript was not revised in line with the comments submitted. There are still errors in the spelling of gene names (e.g. line 293) and methodology. 

The authors need to check the spelling of genes in detail throughout the manuscript and especially the legends of the figures.
Please also send raw data based on which the double delta method was used.

There is information in that manuscript, in the section 4.5: "The relative differences in gene expression were calculated using the 2−ΔΔCt method".
I do not see such data in supplementary files for figures 6-9.
I also do not know what is presented in supplementary materials. There is no description. The numbers are Fold Change or not and what was the true raw data from that experiments (ct ).

Author Response

To Review 1,

My recommendation: accept with minor revision.

 Response 1:

We are grateful to the Reviewer for her/his essential recommendation for our manuscript.

The manuscript was not revised in line with the comments submitted. There are still errors in the spelling of gene names (e.g. line 293) and methodology.

The authors need to check the spelling of genes in detail throughout the manuscript and especially the legends of the figures. 

 Response 2:

We thank this Reviewer for pointing out the issue about the spelling of gene names again and sincerely apologize that we haven't checked this mistake carefully enough. We did this again for the entire manuscript, including the figure legend and methodological section. The issue about line number did not fix between the response comments and the word document may be due to a problem with converting the word file into a PDF file in the MAC system that we used to edit our manuscript. We have tried many times to fix this problem. Unfortunately, we still can not rule it out. Therefore, we recommend the Reviewer use the PDF file when reviewing the manuscript since it is relatively stable in different computer systems. Our apology again for this problematic issue.

Please also send raw data based on which the double delta method was used.

There is information in that manuscript, in the section 4.5: "The relative differences in gene expression were calculated ". 

I do not see such data in supplementary files for figures 6-9.

I also do not know what is presented in supplementary materials. There is no description. The numbers are Fold Change or not and what was the true raw data from that experiments (ct ).

 Response 3:

Many thanks to the Reviewer for her/his vital comments. The qPCR data were presented as fold change compared to the reference cell line using the 2−ΔΔCt method. Following her/his comments, we have included raw data, including CT, ΔCt, ΔΔCt, and 2−ΔΔCt, in supplementary materials. Please find these data on pages 22~27 in the supplemental data file. We also specified this information in the "Materials and Methods” section: “GAPDH was used as reference gene. The relative gene expressions (fold change) were calculated using…". Please find this information in the revised manuscript on lines 678~679 (PDF file).